# Metabolomics as a Tool to Study Underused Soy Parts: In Search of Bioactive Compounds

**DOI:** 10.3390/foods10061308

**Published:** 2021-06-07

**Authors:** Felipe Sanchez Bragagnolo, Cristiano Soleo Funari, Elena Ibáñez, Alejandro Cifuentes

**Affiliations:** 1School of Agricultural Sciences, São Paulo State University (UNESP), Botucatu 18610-034, SP, Brazil; felipe.sanchez@unesp.br (F.S.B.); cristiano.funari@unesp.br (C.S.F.); 2Laboratory of Foodomics, Institute of Food Science Research (CIAL-CSIC), 28049 Madrid, Spain; elena.ibanez@csic.es

**Keywords:** *Glycine max*, foodomics, agricultural waste

## Abstract

The valorization of agri-food by-products is essential from both economic and sustainability perspectives. The large quantity of such materials causes problems for the environment; however, they can also generate new valuable ingredients and products which promote beneficial effects on human health. It is estimated that soybean production, the major oilseed crop worldwide, will leave about 597 million metric tons of branches, leaves, pods, and roots on the ground post-harvesting in 2020/21. An alternative for the use of soy-related by-products arises from the several bioactive compounds found in this plant. Metabolomics studies have already identified isoflavonoids, saponins, and organic and fatty acids, among other metabolites, in all soy organs. The present review aims to show the application of metabolomics for identifying high-added-value compounds in underused parts of the soy plant, listing the main bioactive metabolites identified up to now, as well as the factors affecting their production.

## 1. Introduction

### 1.1. Metabolomics Applied to Agri-Foods and Their By-Products

Plants have been used to produce food, feed, energy, biomaterials, and also as a source of bioactive compounds. Metabolomics has emerged as one of the principal contributors to enhancing the identification of these compounds, generating innovative discoveries and supporting the development of novel products [1]. Progress in efficient extraction techniques, such as ultrasound, microwave, and pulsed-electric-field-assisted extractions, as well as supercritical fluid and pressurized liquid extractions, among others, generate extracts with a higher yield and bioactivity [2,3,4,5,6]. Once these extracts are generated, they can be analyzed with one or more powerful chromatography and/or electrophoresis techniques coupled to high-resolution mass spectrometry (MS) or nuclear magnetic resonance (NMR), producing accurate chemical information on a vast number of compounds [7,8,9,10,11,12]. For the identification of metabolites, databases have been increasingly updated, crosslinking information from different libraries. Sorokina and Steinbeck [13] list almost one hundred databases useful for natural product research. In addition, Global Natural Product Social Molecular Networking (GNPS) and Small Molecule Accurate Recognition Technology (SMART 2.0) are examples of bio-cheminformatics tools for the analysis of MS and NMR data, respectively [14,15,16]. All these modern techniques and tools support the advancement of metabolomics’ frontiers.

In 2019, 8.3 billion metric tons of cereals, oil crops, roots and tubers, sugar crops, and vegetables were produced [17]. However, it is estimated that one-third of food production is lost and wasted, and this problem is Target 12.3 of the 17 Sustainable Development Goals (SDGs) set by the United Nations (UN) [18,19,20]. In this context, foodomics has shown the potential not only of foods, but also of their related by-products, as sources of compounds with human health benefits (Figure 1) [21,22]. For example, Katsinas et al. [23] used supercritical carbon dioxide and pressurized liquid extractions to valorize olive pomace, which is a by-product of the olive oil industry. As a result, they identified several phenolic compounds and generated bioactive extracts. Assirati et al. [24] applied a metabolomics approach in the chemical investigation of the three major solid sugarcane (*Saccharum officinarum*) by-products, leading to the identification of up to 111 metabolites in a single matrix, with several of these compounds already known by their potent bioactive properties, such as 1-octacosanol, octacosanal, orientin, and apigenin-6-C-glucosylrhamnoside. Terpenes of orange (*Citrus sinensis*) juice by-products showed antioxidant and neuroprotective potential in in vitro assays, as revealed by Sánchez-Martínez et al. [25]. As for the permeability of the blood–brain barrier, some terpenes of orange extract demonstrated a high capacity to cross this obstacle, which is a critical point for treating Alzheimer’s disease [25,26].

### 1.2. Glycine max: More Than Beans

Soy, also known as soybean (*Glycine max* (L.) Merr.), is originally from China and Eastern Asia [27]. It is the major oilseed crop worldwide, with a world production of 362, 254, and 61 million metric tons of soy grains, meal, and oil, respectively, in 2020/21. For the same period, the global area harvested was 1.28 million km^2^, 2.5 times the area of Spain [28,29]. Figure 2 shows soybean production from 2000/01 to 2020/21, demonstrating consistent growth, with few moments of decrease [29,30]. However, this production involves just one part of *G. max*: the beans. Krisnawati and Adie [31] analyzed 29 soybean genotypes and found an average value of 1.65 for the straw:grain ratio in soy. Therefore, it is estimated that about 597 million metric tons of soy branches, leaves, pods, and roots will be left on the ground post-harvesting in 2020/21 [29,31]. Figure 3 shows the soil of a no-tillage soybean production, a system which leaves all underused soy parts on the ground. Keeping these materials on the soil contributes to mineral, organic matter, and humidity factors [32]. In contrast, problems related to higher weed and disease infestations, as well as greenhouse gas emissions caused by the decomposition of organic matter, require alternative management of the agricultural straw [33,34,35,36,37,38]. By applying a biorefinery approach, such by-products could be transformed into raw material for the extraction of several bioactive compounds.

Inspired by the potential of underused soy parts, this review aims to show the application of metabolomics in soy analysis, listing the potential of these by-products as a source of high-added-value compounds, as well as the factors which affect their production.

## 2. Metabolomics and Soy

### An Overview

Soy has been recognized as a medicinal plant since it contains several bioactive compounds in its various parts. For example, bioactive peptides found in soybeans have been linked to human health benefits with potential anti-hypertensive, anti-cancer, and anti-inflammatory properties [39]. Another type of bioactive compound identified in soybeans, the anthocyanins, showed anti-obesity and anti- inflammatory properties [40]. Isoflavonoids, the best-known class of compounds found in all parts of soy, have been studied due to their potential protective effects associated with chronic diseases, cancer, osteoporosis, and menopausal symptoms [41,42,43,44,45,46,47].

Different factors modulate a plant’s metabolism, and metabolomics can measure these variations qualitatively and quantitatively, analyzing the production and turnover of primary and secondary (specialized) metabolites [48,49,50]. In soy, metabolomics studies have identified four main causes of changes in metabolism: genetic modifications, organism interactions, growth stages, and abiotic factors. Genetic modifications can be related to different species and cultivar/variety of soybean. Lu et al. [51] investigated the metabolic changes between two soybean species (*Glycine max* and *Glycine soja*) under salt stress. Using gas chromatography coupled to mass spectrometry (GC–MS) and liquid chromatography coupled to Fourier transform and mass spectrometry (LC–FT/MS), the authors found a higher content of hormones, reactive oxygen species, and other substances related to the salt stress condition. In another study, *Glycine max* and *Glycine gracilis* presented different profiles of secondary metabolites during the growth stage, as revealed by a ^1^H NMR-based metabolomics approach [52]. The advancement of molecular biology provides the development of a wide range of soybean cultivars or varieties, with new types of plants resistant against insects, abiotic stress, and other factors. The United States Patent and Trademark Office (USPTO) database reveals 4869 patents for a “soybean cultivar” or “soybean variety” search [53]. Different colored soybeans, such as brown, yellow, or black, present specific metabolite profiles [54,55,56]. Isoflavones could be the substrate for the production of proanthocyanidin in the seed coat, being a possible cause for the brown color of the cultivar Mallikong mutant [55]. Yang et al. [56] identified higher levels of anthocyanin and protein in yellow cotyledon seeds of black soybean. In contrast, higher levels of isoflavone, stearic acid, and polysaccharide are related to the green cotyledon seeds of the same species. Two Korean soy cultivars, Sojeongja and Haepum, presented different levels of soyasaponins Aa and Ab, whose production is related to specific gene variations [57]. Another important factor in genetic modification is the transgenic soybean. García-Villalba et al. [58] used capillary electrophoresis time-of-flight mass spectrometry (CE-TOF-MS) to qualitatively and quantitatively measure the metabolites of transgenic and non-transgenic soybeans. In summary, similar types and amounts of metabolites were identified. The same result was achieved by Harrigan et al. [59] and Clarke et al. [60]. However, it is reported that transgenic soybeans were less affected by generational effects and can present more secondary metabolites, such as prenylated isoflavones [61,62].

Moreover, the interaction between soy and microorganisms, nematodes, aphids, and other insects causes distinct metabolic responses, and metabolomics is a unique approach for understanding such changes, providing insights to improve soy’s response against biotic factors [63,64,65,66,67,68,69,70,71,72,73,74,75,76,77,78,79,80]. Recent works used GNPS to identify metabolite variation in soy infected by the fungus *Phakopsora pachyrhizi* and the nematode *Aphelenchoides besseyi* [78,79,80]. Both pathogens resulted in a higher production of bioactive compounds such as flavonoids, isoflavonoids, and terpenoids.

Distinct metabolic responses have also been reported for each growth stage of soybean [81,82,83]. During germination, 58 metabolites were reported in the separation of soy sprouts, such as phytosterols, isoflavones, and soyasaponins [84]. The production of secondary metabolites such as daidzein, genistein, and coumestrol also changed in the vegetative and reproductive soybean stages, as described by Song et al. [85].

The presence of soybean crops in a wide range of latitudes and longitudes is a consequence of several adaptive changes in their metabolism. Brazil, which is the major producer of soybean, presents different soil and climate types; even so, there is soy production in all its regions. This fact corroborates the high performance of soybean in several abiotic conditions. In addition, treatments with fertilizers and other agricultural inputs have been tested for the cultivation of soybeans in unfavorable conditions, causing additional modifications in soy metabolism [86,87,88,89,90,91,92,93,94]. As an example of external treatments, ethylene application on soybean leaves increased the genistin, daidzin, malonylgenistin, and malonyldaidzin production [94]. Using two ionization methods, electrospray ionization (ESI) and matrix-assisted laser desorption ionization (MALDI), coupled to Fourier transform ion cyclotron resonance-mass spectrometry (FTICR-MS), Yilmaz et al. [95] analyzed the metabolite profile of soy leaves from midsummer to autumn. They found a decreased production of chlorophyll-related metabolites and a higher level of disaccharides from summer to autumn. Another metabolomic approach analyzed soy leaves from crops with different geographical localizations and identified different amounts of metabolites such as pinitol and flavonoids [96]. An excellent review performed by Feng et al. [97] summarizes the use of metabolomics in soy under abiotic stress.

## 3. Bioactive Compounds in Underused Soy Parts

In addition to the four main causes of change in soy metabolism mentioned above, both qualitative and quantitative metabolic variations among soy organs are expected. To present an overview of the metabolite profile of underused soy parts, we selected metabolomics and related works which used various approaches to analyze them [38,67,78,79,80,93,94,98,99,100,101,102,103,104,105,106,107,108]. Using Jchem (JChem for Excel 21.1.0.787, ChemAxon (https://www.chemaxon.com, accessed on 8 April 2021)) [109] and ClassyFire [110], we organized and classified the metabolites identified in soy roots, leaves, branches, and pods, as presented in Appendix A, respectively. Figure 4 summarizes the best-known classes of bioactive compounds identified in underused soy parts. Carboxylic acids and their derivatives, such as amino acids, peptides, and analogues, are the most mentioned class of compounds. This class is mainly composed of primary metabolites; however, it also contains several bioactive compounds. Similarly, organooxygen and fatty acyl compounds include metabolites with human health benefits. Isoflavonoids, which are the most mentioned class of secondary metabolites, as well as prenol lipids and flavonoids, have been suggested to have a wide range of medicinal uses. Focusing on secondary metabolites, prenol lipids are the most identified class of compounds in soy roots, with several soyasaponins found in this part. In soy leaves, different subclasses of isoflavonoids have been found, such as isoflavonoid O-glycosides, isoflavans, isoflav-2-enes, and others. The metabolite profiles of soy branches and pods have been less studied; however, approximately 20 flavonoids and isoflavonoids have been identified in each part. Other classes of compounds, such as steroids and steroid derivatives, coumarins and derivatives, and cinnamic acids and derivatives, have been found in underused soy parts.

Table 1 presents 38 isoflavonoids identified in one or more of the above-mentioned underused soy parts. Eight of them (daidzein, genistein, glycitein, daidzin, genistin, glycitin, malonyldaidzin, and malonylgenistin) were reported in all soy organs. Recent works showed promising biological activities of daidzein against colon cancer and hepatitis C virus [111,112]. Daidzin, which is a glyco-conjugate form of daidzein, presented therapeutic properties against multiple myeloma and epilepsy [113,114]. Bioactivity studies regarding the other aforementioned compounds also found properties against chronic vascular inflammation, human gastric cancer, breast cancer, and degenerative joint diseases [115,116,117]. Biochanin A, coumestrol, glyceollin, medicarpin, and ononin are more examples of widely known bioactive isoflavonoids which are found in different soy organs (see Table 1 for a summary) [118,119,120,121,122]. Carneiro et al. [38] quantified six isoflavones in soy branches, leaves, pods, and beans collected just after mechanical harvesting. Almost 3 kg of isoflavones were found per metric ton of soy leaves. However, less than 1 kg per metric ton was found in soy branches and pods. In soybeans, which are the main product of the soy plant, it was approximately 2 kg per metric ton.

### 3.1. Roots

Different compounds belonging to the prenol lipids category, which are recognized by their bioactivity, have already been identified in soy. Appendix A contains 339 compounds found in soy roots, of which 33 are of this class [79,93,98,101,103,106,108]. Tsuno et al. [108] identified several soyasaponins, sapogenins, and isoflavones in soy root exudates. Soyasaponins have been linked to anti-obesity, anti-oxidative stress, and anti-inflammatory properties, as well as preventive effects on hepatic triacylglycerol accumulation [123,124,125,126]. Omar et al. [127] identified the potent inhibitory effects of soyasapogenol A, which is a triterpenoid, against p53-deficient aggressive malignancies. In addition, other compounds of different classes, such as fatty acyls, isoflavonoids, flavonoids, and others, are presented in Appendix A. Linoleic acid, naringenin, and formononetin-7-O-glucoside, which are examples of the aforementioned classes, have been related to cardiovascular health, neuroprotective effects, and anti-inflammatory properties [122,128,129]. The chemical structures of these bioactive compounds are presented in Figure 5.

### 3.2. Leaves

Leaves and roots are the most-studied underused soy parts. In Appendix A, 259 metabolites of 32 classes identified in soy leaves are presented [38,78,80,94,98,101,102,103,106]. Almost 90 of these compounds are flavonoids, isoflavonoids, or prenol lipids. Widely known bioactive flavonoids such as apigenin, kaempferol, rutin, and others were also identified. Apigenin has been suggested as a potential anticancer agent [130]. Glyceollin I and soyasaponin I, an isoflavonoid and a prenol lipid, presented activities against breast cancer and Parkinson’s disease, respectively [120,131]. Moreover, different soyasaponins and even trigonelline, which is an alkaloid, were found in this part of the plant. For example, the latter substance was reported to have potential for lung cancer therapy, memory function recovery, and an anti-obesity effect [132,133,134]. Figure 6 shows the chemical structures of the aforementioned metabolites.

### 3.3. Branches

In soy branches, 197 compounds have already been identified, as presented in Appendix A [38,67,98,99,101]. The most widely reported class among these metabolites is the organooxygen compounds category (53 compounds), such as alcohols and polyols, carbohydrates and their conjugates, and carbonyl (Appendix A). Shikimic acid, an example of an organooxygen compound, was linked to therapeutic effects in osteoarthritis [135]. Metabolites of other classes, such as succinic and stearic acids, presented an apoptotic effect in T-cell acute lymphoblastic leukemia and antifibrotic activity, respectively [136,137]. Flavonoids and isoflavonoids, such as 7,4′-dihydroxyflavone and glycitin, presented activity against lung diseases [138,139]. The chemical structures of these compounds are shown in Figure 7.

### 3.4. Pods

Similarly to branches, there are few metabolomics works identifying pod metabolites [38,98,100,104,105,107]. Amino acids, peptides, and mono-, di-, and tricarboxylic acids and their derivatives are the most mentioned types of compounds in pods, as shown in Appendix A, with some of these substances already widely used in industry, such as citric and fumaric acids. Moreover, specialized metabolites such as camphene and α-pinene, which were also identified in soy pods, presented anti-skeletal muscle atrophy and neuroprotective effects, respectively [140,141]. Quercetin, which is a widely known flavonoid, may be a potential anti-inflammatory treatment in patients with COVID-19, as described by Saeedi-Boroujeni and Mahmoudian-Sani [142]. Hexadecanoic acid, a fatty acyl compound, presented an inhibitory effect on HT-29 human colon cancer cells [143]. Figure 8 presents the chemical structures of one compound of each class mentioned. In addition, fatty acyls, flavonoids, isoflavonoids, and other classes of compounds were identified in pods, leading to the 94 metabolites presented in Appendix A.

## 4. Bioactive Compounds in Industrial By-Products from Soybean Processing—An Overview and Trends

Underused soy parts are not the only by-products of soy. Soybean is transformed into different products, such as flour, oil, tofu, soy sauce, and soy milk, as presented in Figure 9. These processes generate by-products with vast applications, and their use as sources of bioactive compounds is an excellent opportunity to develop new ingredients and products with health benefits. In addition, green extractions of such materials provide more sustainable and valuable outcomes. Soy hull represents approximately 8% of the bean and is one of the by-products generated by soy flour and oil production [144]. Using a sustainable approach for valorizing soybean hull, Cabezudo et al. [145] optimized alkaline hydrolysis for polyphenol extraction and tested a fermentation with *Aspergillus oryzae* and α-amylase hydrolysis for the same purpose. In this work, phenolic acids, anthocyanins, and isoflavones were identified by LC–MS, demonstrating the great potential of soy hull as a source of antioxidant compounds. Soybean meal is a by-product of soy oil production. The characterization of soybean meal demonstrated a higher antioxidant capacity and content of total phenolics, flavonoids, and saponins than in unprocessed grains [146]. In addition, different forms of isoflavones, such as β-glucosylated, malonyl glucosylated, acetyl glucosylated, and aglycones, as well as three group B soyasaponins were identified. Alvarez et al. [147] optimized a green supercritical fluid extraction using CO_2_ and ethanol to analyze soybean meal, resulting in extracts with antioxidant properties and a higher content of phenols and flavonoids. Freitas et al. [148] also used green extraction to obtain an aqueous extract of soy meal with a high inhibition of lipid peroxidation, identifying 16 phenolics in the extract. Okara and soy whey, which are by-products of soymilk and tofu production, have also been used as a source of bioactive compounds. Nkurunziza et al. [149] extracted different isoflavone aglycones of okara using subcritical water. In addition, Nile et al. [150] found a high level of isoflavones in okara, as well as potential antioxidant, anti-inflammatory, and inhibitory enzyme activities. Liu et al. [151] used foam fractionation and acidic hydrolysis to remove proteins from soy whey, generating extracts with a high level of isoflavone aglycones. Other uses for and information about soy whey are described by Chua and Liu [152] and Davy and Vuong [153].

Additionally, biotechnology advances produce new opportunities to develop fermented soy products with more health benefits [154,155,156,157]. Furthermore, metabolomics contributes to a better understanding of the biotransformation of bioactive compounds [158]. As a substrate for microorganisms, okara has been widely used in different fermented modes [159,160,161,162,163,164]. *Rhizopus oligosporus*, *Bacillus subtilis* WX-17, and *Eurotium cristatum* were used for okara fermentation, resulting in products with higher bioactivity, nutritional composition, and anti-diabetic potential [159,160,161,162,163,164]. In addition, higher levels of phenolics, flavonoids, and biotransformed substances were identified. The fermentation of other soy by-products, such as meal, whey, and hull, has demonstrated vast potential for enhancing the bioactivity of the final products [165,166,167,168,169,170,171].

## 5. Conclusions and Outlook

Soy is the major oilseed crop worldwide, and its large production generates a massive amount of by-products. The presence of isoflavonoids, flavonoids, terpenes, and other substances in soy branches or stems, leaves, pods, and roots, provides insights into the use of these underused materials as a source of bioactive compounds. In contrast, challenges for the valorization of such by-products remain. A multi-omics approach, as proposed by foodomics, may reduce the gap between the crude parts of soy and the final ingredient or product, increasing the safety and quality of all the processes and products involved. Moreover, soy metabolomics studies have focused on specific organs as well as metabolic modifications, resulting in an incomplete metabolite profile of all soy parts. In summary, our review demonstrates the extensive use of metabolomics in soy research and how this work provides new information for alternative uses of underused soy parts with more added value. Interestingly, our work also shows that there are many (underused) soy compounds that still need to be interrogated for their potential bioactivity and possible health benefits.

In addition, more studies about the life-cycle assessment (LCA) of the soybean supply chain are required to analyze potential problems related to the high amount of by-products which are left on the ground post-harvest. Environmental problems such as soil and water contamination could occur due to the presence of several bioactive compounds present in such agricultural by-products. The use of green extraction and biotechnology could be feasible alternatives for the re-use of these materials.

## Figures and Tables

**Figure 1 foods-10-01308-f001:**
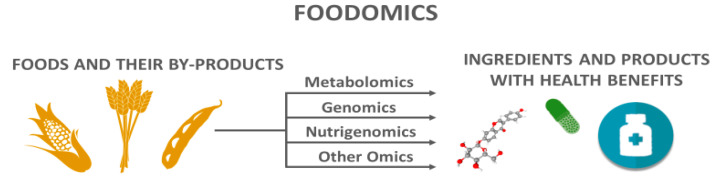
Foodomics proposes a holistic approach to develop ingredients and products with health benefits from foods and their by-products.

**Figure 2 foods-10-01308-f002:**
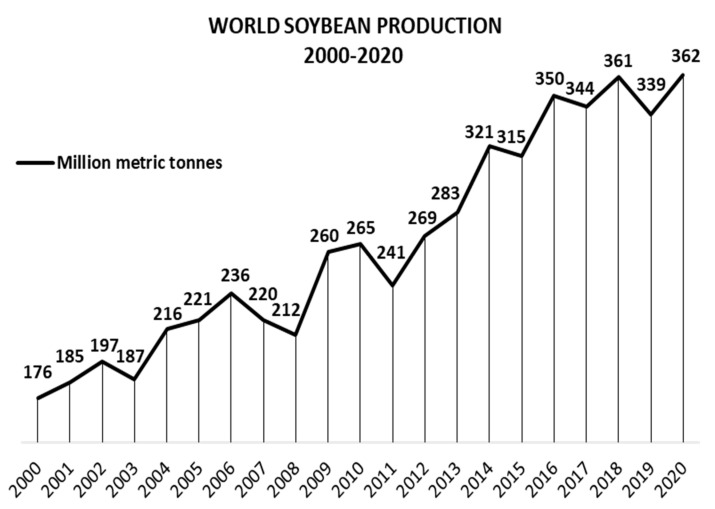
World soybean production 2000–2020, in million metric tons.

**Figure 3 foods-10-01308-f003:**
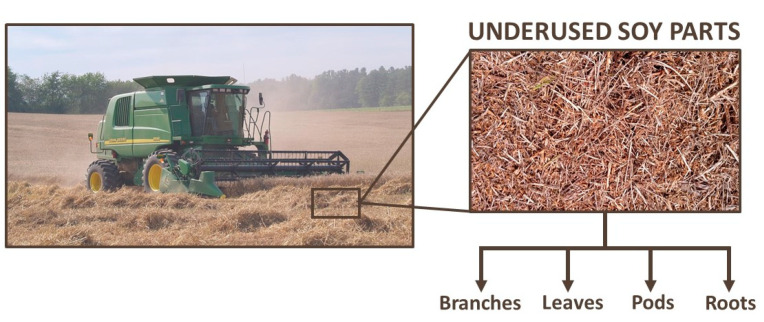
Underused soy parts left on the soil just after the soybean harvest.

**Figure 4 foods-10-01308-f004:**
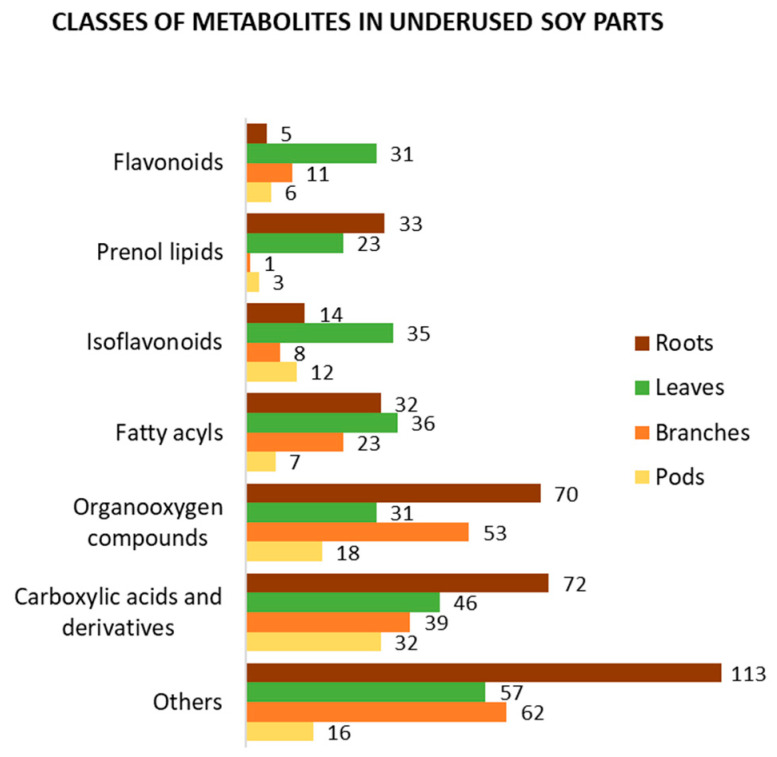
Classification of the metabolites identified in soy roots, leaves, branches, and pods according to ClassyFire.

**Figure 5 foods-10-01308-f005:**
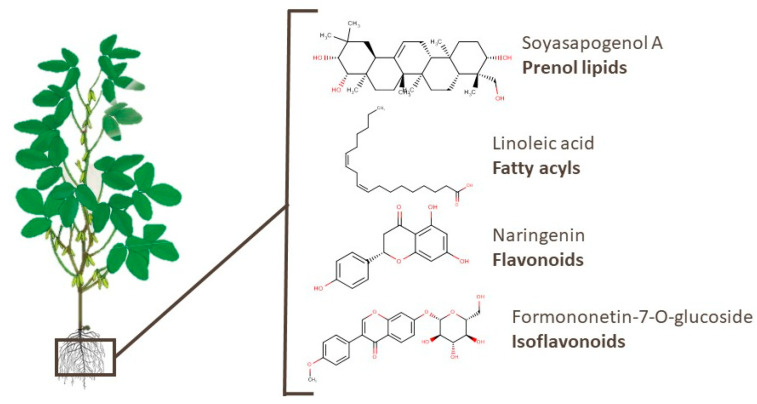
Chemical structures of soyasapogenol A, linoleic acid, naringenin, and formononetin-7-O-glucoside, which are examples of bioactive compounds identified in soy roots.

**Figure 6 foods-10-01308-f006:**
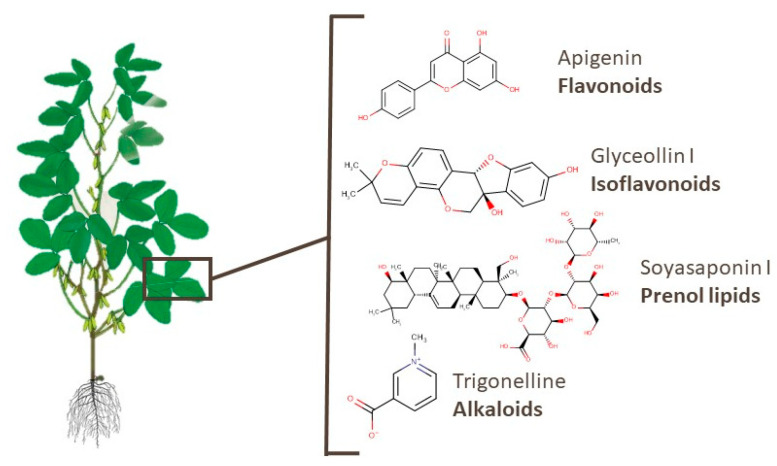
Chemical structures of apigenin, glyceollin I, soyasaponin I, and trigonelline, which are examples of bioactive compounds identified in soy leaves.

**Figure 7 foods-10-01308-f007:**
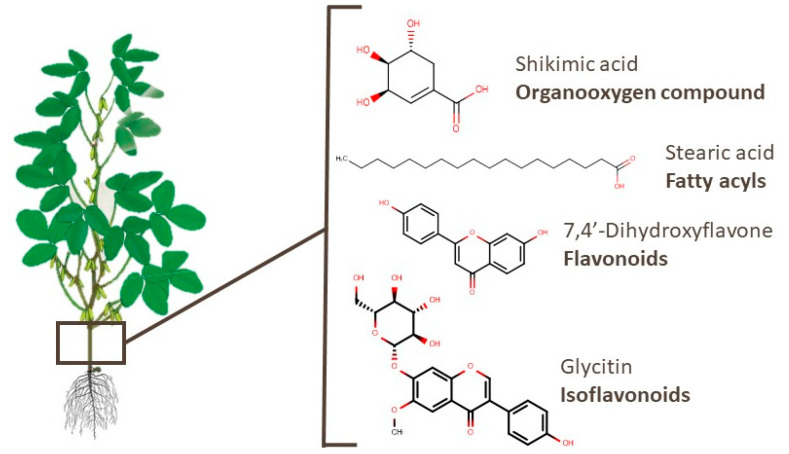
Chemical structures of shikimic acid, stearic acid, 7,4′-dihydroxyflavone, and glycitin, which are examples of bioactive compounds identified in soy branches.

**Figure 8 foods-10-01308-f008:**
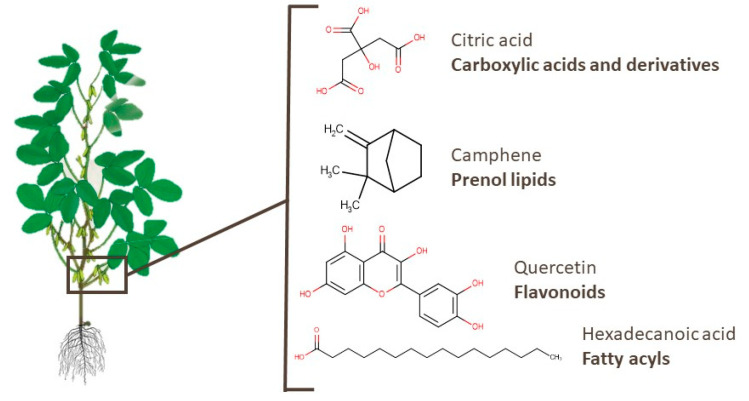
Chemical structures of citric acid, camphene, quercetin, and hexadecanoic acid, which are examples of bioactive compounds identified in soy pods.

**Figure 9 foods-10-01308-f009:**
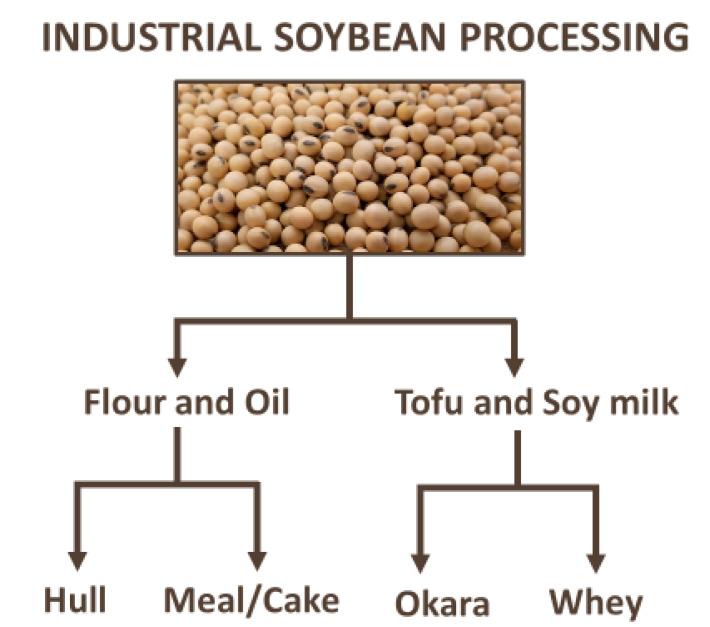
Industrial soybean processing and its respective products and by-products.

**Table 1 foods-10-01308-t001:** Isoflavonoids identified in soy branches (B), leaves (L), pods (P), and roots (R).

Name	Formula	B	L	P	R	References
2′-hydroxydaidzein	C_15_H_10_O_5_		X			[80]
7,3′,4′-trihydroxyisoflavone	C_15_H_10_O_5_				X	[79]
7-O-methylluteone	C_21_H_20_O_6_		X			[78]
acetyl daidzin	C_22_H_22_O_9_			X		[104]
acetyl genistin	C_23_H_22_O_11_		X	X		[94,104]
acetyl glycitin	C_24_H_24_O_11_			X		[104]
afrormosin 7-O-glucoside	C_23_H_24_O_10_		X			[80]
biochanin A	C_16_H_12_O_5_		X			[80]
biochanin A 7-O-D-glucoside	C_22_H_22_O_10_		X			[80]
biochanin A 7-O-glucoside-6′′-O-malonate	C_25_H_24_O_13_		X			[80]
calycosin	C_16_H_12_O_5_		X			[80]
coumestrol	C_15_H_8_O_5_		X		X	[79,80,101]
daidzein	C_15_H_10_O_4_	X	X	X	X	[38,78,79,80,94,98,101,103,104,106,107,108]
daidzin	C_21_H_20_O_9_	X	X	X	X	[38,78,79,80,94,98,101,103,104,107,108]
formononetin	C_16_H_12_O_4_		X			[80,102]
formononetin 7-O-glucoside	C_22_H_22_O_9_		X		X	[79,80]
formononetin 7-O-glucoside-6′′-malonate	C_25_H_24_O_12_		X			[78,80,94]
formononetin 7-O-glucoside-6-O-malonate	C_25_H_24_O_12_		X		X	[78,79]
genistein	C_15_H_10_O_5_	X	X	X	X	[38,79,94,98,104,108]
genistin	C_21_H_20_O_10_	X	X	X	X	[38,78,79,94,101,104,107,108]
glyceollidin I/II	C_20_H_20_O_5_		X			[80]
glyceollin I	C_20_H_18_O_5_		X			[78,80]
glyceollin II	C_20_H_18_O_5_		X			[78,80]
glyceollin III	C_20_H_18_O_5_		X			[78,80]
glyceollin IV	C_21_H_22_O_5_		X			[80]
glyceollin VI	C_20_H_16_O_4_		X			[80]
glycitein	C_16_H_12_O_5_	X	X	X	X	[38,80,98,104,108]
glycitein 7-O-glucoside	C_22_H_22_O_10_		X			[80]
glycitin	C_22_H_22_O_10_	X	X	X	X	[38,79,101,104,108]
isotrifoliol	C_16_H_10_O_6_		X			[80]
malonyldaidzin	C_24_H_22_O_12_	X	X	X	X	[38,78,79,80,94,101,103,104,107,108]
malonylgenistin	C_24_H_22_O_13_	X	X	X	X	[78,79,80,94,101,104,107,108]
malonylglycitin	C_25_H_24_O_13_		X	X	X	[80,94,104,108]
medicarpin	C_16_H_14_O_4_		X			[80]
neobavaisoflavone	C_20_H_18_O_4_		X		X	[78,79]
phaseollin	C_20_H_18_O_4_		X			[80]
pisatin	C_17_H_14_O_6_		X			[80]
sojagol	C_20_H_16_O_5_		X			[78,80]

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
