# Peer review of "Metabolomics as a Tool to Study Underused Soy Parts: In Search of Bioactive Compounds"

_foods, 2021, doi:10.3390/foods10061308_

Round 1

Reviewer 1 Report

The review of Bragagnolo et al. about "Metabolomics as a tool to study underused soy parts: in search of bioactive compounds" is a good piece of work, comprehensive and well written. Besides being a review of a large number of research works on the metabolomic approach applied to the topic, it also binds together metabolites belonging to different anatomic parts of the whole soy plant, making feasible a deeper consideration on the potential of processing soy-related by-products, both those left on the ground and those originating from industrial transformation processes, towards the recovery of valuable bioactive compounds.

In my opinion the only weak section of the review is Section 4 and somehow also the Conclusions. I would have expected a deeper analysis of the economical convenience of treating such by-products to answer the questions: "is there, currently, an economical convenience on treating such a large amount of by-products of the soy production chain? Are there green methods that are also economically sustainable or is it likely that they can be developed in a forthcoming future?". A sort of overall concise Life Cycle Assessment (LCA) would have improved the value of the review greatly, in my opinion. If the authors do not feel confident on adding such important information in Section 4, I would however expect them to consider the issue in the Conclusions, strengthening the relevance of their work.

Some minor issues:

Page 1, line 30: Change "were" with "have been" or "are". The simple past does not fit here.

Page 2, Figure 3: The figure, which is a close-up detail of a soy field after harvest, in itself is not very informative. I think it would be better to show the harvested field in a larger view and insert the same figure as an inset (zoom in) to actually show what remains on the soil after harvest.

Page 5, line 161: I do not understand the use of "were informed" here. I think something like "were presented" or "were discussed" would fit better.

The figures, in general, do not add much value to the review. The authors might consider whether they want to improve them by either adding visual information or condensing them into a larger figure depicting the whole flow from by-products to bioactive metabolites.

Author Response

Reviewer 1 of the manuscript titled “Metabolomics as a tool to study underused soy parts: in search of bioactive compounds”.

01st June, 2021

Point 1: The review of Bragagnolo et al. about "Metabolomics as a tool to study underused soy parts: in search of bioactive compounds" is a good piece of work, comprehensive and well written. Besides being a review of a large number of research works on the metabolomic approach applied to the topic, it also binds together metabolites belonging to different anatomic parts of the whole soy plant, making feasible a deeper consideration on the potential of processing soy-related by-products, both those left on the ground and those originating from industrial transformation processes, towards the recovery of valuable bioactive compounds. In my opinion the only weak section of the review is Section 4 and somehow also the Conclusions. I would have expected a deeper analysis of the economical convenience of treating such by-products to answer the questions: "is there, currently, an economical convenience on treating such a large amount of by-products of the soy production chain? Are there green methods that are also economically sustainable or is it likely that they can be developed in a forthcoming future?". A sort of overall concise Life Cycle Assessment (LCA) would have improved the value of the review greatly, in my opinion. If the authors do not feel confident on adding such important information in Section 4, I would however expect them to consider the issue in the Conclusions, strengthening the relevance of their work.

Response 1: Dear Reviewer 1, thank you for your kind words and suggestions. First, we added more information about the by-products of industrial soybean processing, providing a wide range of potential uses of such materials. Second, we preferred to discuss ¡the Life Cycle Assessment (LCA) of the soybean supply chain in the Conclusions. In this part, we provided an insight about the application of LCA focused on the bioactive compounds present in the underused soy parts, which are left on the ground and could cause environmental problems.

Point 2: (Minor issues)

Page 1, line 30: Change "were" with "have been" or "are". The simple past does not fit here.

Response: Thank you. It has been corrected in the new version.

Page 2, Figure 3: The figure, which is a close-up detail of a soy field after harvest, in itself is not very informative. I think it would be better to show the harvested field in a larger view and insert the same figure as an inset (zoom in) to actually show what remains on the soil after harvest.

Response: Thank you. It has been corrected in the new version.

Page 5, line 161: I do not understand the use of "were informed" here. I think something like "were presented" or "were discussed" would fit better.

Response: Thank you. It has been corrected in the new version.

The figures, in general, do not add much value to the review. The authors might consider whether they want to improve them by either adding visual information or condensing them into a larger figure depicting the whole flow from by-products to bioactive metabolites.

Response: Thank you. It has been corrected, we improved the Figures in the new version.

Reviewer 2 Report

Even if the existence of bioactive substances is known by metabolomics, the content is an important index for effective utilization. In order to extract from the unused part of soybean, the concentration of bioactive substances must be known. There is no utility value at low concentrations of bioactive substances. Whenever possible, indicate the concentration of bioactive substances.

Author Response

Reviewer 2 of the manuscript titled “Metabolomics as a tool to study underused soy parts: in search of bioactive compounds”.

01st June, 2021

Point 1: Even if the existence of bioactive substances is known by metabolomics, the content is an important index for effective utilization. In order to extract from the unused part of soybean, the concentration of bioactive substances must be known. There is no utility value at low concentrations of bioactive substances. Whenever possible, indicate the concentration of bioactive substances.

Response 1: Dear Reviewer 2, thank you for your suggestions. We agree with you that it would be great to provide some discussion about the content of bioactive compounds in soy parts. We added one specific paper [Food Research International 2020, 130, doi:10.1016/j.foodres.2019.108949] that already analyzed the content of isoflavones in soy agro by-products. However, there are few works which analyzed the content of other bioactive metabolites, and our contribution with our paper was to show that there is a considerable number of high added-value compounds in the underused soy parts, providing insights for new studies to quantify specific compounds of biomedical interest and others industrial applications.

In addition, we added more information about the bioactive metabolites identified in different soy organs and by-products of industrial soybean processing; improved the Figures; and added a short conclusion about the Life Cycle Assessment (LCA) in soybean supply chain.
